# Heart Rate Variability Responses to a Training Cycle in Female Youth Rowers

**DOI:** 10.3390/ijerph17228391

**Published:** 2020-11-13

**Authors:** Rohan Edmonds, Julian Egan-Shuttler, Stephen J. Ives

**Affiliations:** 1Department of Exercise Science and Pre-Health Professions, Creighton University, Omaha, NE 68178, USA; rohanedmonds@creighton.edu; 2Health and Human Physiological Sciences Department, Skidmore College, Saratoga Springs, NY 12866, USA; jeganshu@skidmore.edu; 3School of Sport and Exercise Sciences, Loughborough University, Leicestershire LE11 3TU, UK

**Keywords:** autonomic nervous system, cardiac autonomic, vagal tone, adolescents, training load

## Abstract

Heart rate variability (HRV) is a reputable estimate of cardiac autonomic function used across multiple athletic populations to document the cardiac autonomic responses to sport demands. However, there is a knowledge gap of HRV responses in female youth rowers. Thus, the purpose of this study was to measure HRV weekly, over a 15-week training period, covering pre-season and up to competition in youth female rowers, in order to understand the physiological response to long-term training and discern how fluctuations in HRV may relate to performance in this population. Measures of heart rate and heart rate variability were recorded before training each Friday over the monitoring period in seven athletes. Analysis of heart rate variability focused on time domain indices, the standard deviation of all normal to normal R–R wave intervals, and the root mean square of successive differences as markers of cardiac parasympathetic modulation. Training load was quantified by multiplying the rating of perceived exertion of the weeks training and training duration. A decrease was identified in cardiac parasympathetic modulation as the season progressed (Effect Size (Cohen’s *d*) = −0.34 to −0.8, weeks 6 and 11–15), despite no significant relationship between training load and heart rate variability. Factors outside of training may further compound the reduction in heart rate variability, with further monitoring of external stressors (e.g., school) in adolescent athletes.

## 1. Introduction

In a sustained effort to maximize human performance, coaches and sports scientists aim to understand how individual athletes respond to training. If training is effectively monitored, programs may be adjusted to prevent training maladaptation and ultimately, improve performance [1]. An assessment that has shown potential in monitoring training and maladaptation in athletic populations is the measurement of heart rate variability [HRV] [2,3], the non-invasive estimate of cardiac autonomic nervous system activity at rest [4,5] and following exercise [6,7,8]. 

Heart rate variability may represent the fine tuning of impulses from the sympathetic and parasympathetic nervous systems [4,9], and has been shown to reflect training load in athletes [10,11]. While not always the case [12,13], vagal indices of HRV are often reduced during periods of intense training [11,14,15]. Accordingly, HRV has been utilized to non-invasively monitor fluctuations associated with altered cardiac autonomic activity in athletic populations during seasonal training. Oliveira et al. measured the HRV of male futsal players before a pre-season training period, after pre-season, and during their competitive season, finding that HRV was increased due to training, and this increase was reflected by performance measures [16]. In addition, Edmonds et al. conducted weekly HRV measurements within a cohort of Paralympic swimmers over a 14-week training and competitive season [17]. Although Edmonds et al. [17] found that HRV was relatively stable during the athletes training cycle, only reducing following an early season competition, all athletes recorded personal best times, a potential indication that training workload was well managed. This also potentially eludes to the notion that stable HRV is favorable, suggesting that weekly measurement of HRV may be useful in different athletic populations.

It is also valuable for coaches and sports scientists to assess an athlete’s readiness to perform, with increased estimates of cardiac parasympathetic nervous system activity associated with increased athletic performance [2,18,19]. Previous research in swimmers has shown that increased estimates of parasympathetic activity following a taper correlated highly with increased swimming performance [18,19]. Comparable results were found by Cataldo et al. in youth male soccer players, as higher peak power during repeated sprint exercise was highly correlated with indicators of parasympathetic activity [20]. Research has also identified negative correlations between vagal indices of HRV following training and performance, with lower parasympathetic modulation (i.e., increased sympathetic modulation) shortly following training associated with improved performance [21]. These findings suggest that HRV could also be used to evaluate whether an athlete is primed for performance; however, little work has been done on youth athletes and females in particular.

Currently, HRV has been researched in a variety of elite sports [2,22,23], yet rowing—another sport that requires a high training load and intensity—has been sparingly investigated. An early long-term study of male Italian Olympic rowers, while potentially limited by its sole analysis of frequency domain measures, reported that changes in training load affect HRV, likely through variations in cardiac sympathetic activation over a short period (~20 days) [24]. More recently, a study indicated a relationship between training load and HRV in male and female Olympic rowers, showing that periods of training with increased intensity suppress parasympathetic activity, while periods of low-intensity training preserve, and often increase, vagal activity [25]. Again, in elite male rowers, alterations in heart rate variability leading up to world championships suggested athlete-dependent effects, but 3 of 4 athletes displayed an increase in estimated cardiac parasympathetic activity [26]. However, these investigations evaluated exclusively elite, mostly male, rowers with advanced coaches, sports scientists, and monitoring, potentially limiting its applicability to amateur, and/or youth athletes. Indeed, research in youth athletes without such resources, specifically females, has been limited to date. To this end, recent work in youth female rowers explored the impact of travel to, and response to, a week-long intense training camp, and found that travel and the increased training load acutely reduced HRV [27]. Although, much less is known about how long-term training influences cardiac autonomic activity in these young female rowing athletes, and how HRV may or may not relate to rowing performance.

As such, the purpose of this study was to measure HRV, once a week, during a 15-week training period, from pre-season up to the competition phase, to provide insight into the physiological responses to long term training in high performing youth female rowers. Additionally, we sought to examine the relationship between cardiac autonomic function and rowing performance (2000 m ergometer time trial).

## 2. Materials and Methods

### 2.1. Participants

Participants in the study were recruited from the largest team in a 100-mile radius (~180 athletes). To best minimize confounding factors, such as sex, age, and training age, paired with the lack of studies in females, recruitment was limited to experienced varsity female rowers. To maintain homogeneity, participants were recruited from a pool of twenty-five female varsity rowers who were members of a local competitive high school rowing program. This program had consistently won or challenged for national titles. Seven rowers (Table 1) enrolled in the study and were on average 16.6 (±1.0) years old, with an average height of 173.9 (±7.6) cm and weight of 67.6 (±10.3) kg. The rowers were all experienced, training six or more times a week, and had been rowing competitively for at least three seasons. The average 2000 m ergometer performance times for these athletes was 468 ± 28 s, which placed these athletes within a “high performance” level of rowing ability, when compared against US rowing junior performance standards. Written informed assent was obtained from all participants and their legal guardians prior to participation in this study. This study was approved by the local Institutional Review Board (IRB# 1605-515) in accordance with the ethical standards set forth in the Declaration of Helsinki.

### 2.2. Experimental Design

Heart rate recordings took place once each week over a 15-week winter training season, which included transitions from pre-season to in-season to competition preparation, culminating in the spring racing season. The first race of the season occurred the weekend after the last week of training (week 15). The measurement from the first week of training was classified as a baseline recording, taken as described below. During week 6 of the training period, there was a week-long training camp in which training frequency was increased (10 sessions per week compared to 6), and the HRV response to such overload was described previously in a similar population [27]. All HRV measurements were recorded once a week, on the Friday of each training week, and at a consistent time of day (3:00 p.m.) to eliminate any circadian effect on HRV [28,29]. While daily HR recording within elite athlete populations is considered the “gold standard” for HRV assessment [1], research has also shown similar weekly HRV assessment may be all that is necessary in highly trained athletes [17,30], and even daily measures are often reported by week in longer-term studies [26]. Given issues with the practicality of daily HRV assessment, an afternoon, single pre-training recording was used in order to assess the athlete’s state immediately prior to their last session of the week. Heart rate was recorded over a 10-min period using the Zephyr BioHarness (Zephyr Technology, Annapolis, MD, USA) with recordings sampled at 1000 Hz for determination of RR intervals. In line with previous studies, the first five minutes of monitoring was used as a stabilization period to ensure athletes were rested, with HRV calculated from the final five minutes [31,32]. To maintain uniformity, HR measurements were obtained in a seated position [3], and breathing was spontaneous [32]. Rating of perceived exertion for the week of training was measured before every HR recording using the modified Borg’s scale (RPE; 1–10) [33], and was used to calculate training load, as originally described [34]. This method has since been used in similar studies [27], and has been systematically reviewed, supporting its use as a standalone approach [35].

Three 2000 m rowing performance tests were also performed during the training period, at the end of weeks 4, 8, and 12 (Table 2). Rowing performance tests were also performed on Fridays, importantly, only after HRV recordings. The gold standard performance test in rowing (2000 m time trial) was performed on the Concept2 rowing ergometer (Model D, Concept2, VT, USA) in accordance with Junior National team suggested guidelines (e.g., drag factor, etc.) and requirements (e.g., open rating, stationary, and confirmed by coach).

### 2.3. Data and Statistical Analysis

All RR interval recordings were analyzed using Kubios HRV software (v2.2, University of Kuopio, Kuopio, Finland) with HRV calculated from the last 5 min of the 10-min recording period based on previously established guidelines (Task Force, 1996). Prior to analysis, any artifact or ectopic beats were corrected via Kubios’ in-built cubic spline interpolation [36] with an artifact correction threshold set at very low. Coupled with mean HR, analysis of HRV included the standard deviation of all normal to normal RR intervals (SDNN), a marker of global autonomic nervous activity at the heart, and the root mean square of successive differences (RMSSD), which is thought to represent parasympathetic influence. Previous research has used both RMSSD and SDNN as estimates of cardiac autonomic activity in athletic populations [2,37,38,39].

Given the small sample size, meaningful changes from pre-training camp (baseline) values were assessed using magnitude-based decisions (MBD) [40]. In line with previous research, magnitude-based decisions were undertaken by defining the smallest worthwhile change from baseline (±3%) [31]. In line with a previously outlined framework [40], the chances of meaningful negative, trivial, or positive changes were determined qualitatively as follows: almost certainly (99.5%); very likely (99.5–95%); likely (95–75%); possibly (75–25%); unlikely (25–5%); very unlikely (5–0.5%); almost certainly not (<0.5%). If the chances of having a positive and negative change were both greater than 5%, the true difference was deemed unclear. In line with previous research, the observed changes were standardized following Cohen’s effect size principle (i.e., changes in the mean divided by the between-athlete SD of baseline data) [31]. The magnitude of the weekly change from baseline (pre-season) during the training camp was determined using Cohen’s *d* (i.e., Effect Size, ES) with threshold values established as small (0.2), moderate (0.6), large (1.2), and very large (2.0) [40]. Differences in performance (2000 m time trial) over time were assessed using a one-way analysis of variance, with an alpha level of *p* < 0.05 and times expressed as mean (± SD). In addition, differences in mean HR, HRV, training load, and rowing performance were examined using a repeated measures analysis of variance (ANOVA), with a Bonferroni post hoc test, to identify potential differences over the 15-week monitoring period, with an alpha level established at *p* < 0.05 for all analyses.

Pearson correlations were also examined using the Statistical Package for Social Sciences (SPSS) software (v23, SPSS INC., Chicago, WI, USA) to identify relationships, using individual data, between mean HR, HRV, and training load, at each rowing time trial timepoint (week 4, 8, and 12).

## 3. Results

### 3.1. Training Load

Training load was likely higher in week 6 (ES = 0.54) (Table 3, Figure 1A). Additionally, training load was very likely higher in week 12 (ES = 1.13) compared to baseline (Figure 1A). In contrast, training load was likely lower in week 5 (ES = −0.50) compared to baseline (Figure 1A).

Training load at week 12 (2286.0 ± 219.9) was significantly higher compared to week 5 (1520.0 ± 474.3, *p* = 0.04, *d* = −2.65) and week 8 (1425.0 ± 329.1, *p* = 0.02, *d* = −2.96). There were no other differences reported for training load over the monitoring period.

Over the monitoring period, athlete #3 (2029.5 ± 519.4) reported a significantly higher training load when compared to compared to athlete #4 (1415.5 ± 435.2, *p* = 0.02, *d* = 1.03) and athlete #5 (1460.1 ± 354.4, *p* = 0.006, *d* = 1.23). Training load was similar between all other athletes over the monitoring period.

There were no correlations observed between training load and mean HR, RMSSD, or SDNN at any time point during the monitoring period (all *p* > 0.05).

### 3.2. Heart Rate and Heart Rate Variability

Mean HR was likely higher at week 12 (ES = 0.66) compared to baseline (Figure 1B, Table 3). Excluding week 12, variation in mean HR across the monitoring period was unclear (Table 3).

A significant difference (*p* = 0.047, *n*^2^ = 0.234) in mean HR was observed over the 15-week monitoring period; however, the Bonferroni post hoc test was unable to identify where that difference was. Athlete #7 (65.6 ± 7.8 bpm) had a significantly lower (*p* < 0.01) mean HR compared to all other athletes during monitoring period.

RMSSD was possibly lower at week 13 (ES = −0.38) compared to baseline (Table 3, Figure 1C). Additionally, RMSSD was likely lower at week 6 (ES = −0.34), week 14 (ES = −0.38), and week 15 (ES = −0.35) compared to baseline (Table 3, Figure 1C). Finally, changes in RMSSD were very likely lower at week 11 (ES = −0.70), and week 12 (ES = −0.80) compared to baseline (Table 3, Figure 1C). Interestingly, RMSSD was very likely higher at week 13 (ES = 0.51), and likely higher at week 14 (ES = 0.45) and week 15 (ES = 0.48) when compared to week 12 (its lowest over the 15 weeks) (Figure 1C).

A significant difference (*p* = 0.03, *n*^2^ = 0.247) in RMSSD was reported during the 15-week monitoring period; however, the Bonferroni post hoc test was unable to identify where the difference was. Significant differences in RMSSD were also observed between athletes over the 15-week monitoring period. Athlete #1 (49.4 ± 28.2 ms) reported a significantly lower RMSSD when compared to athlete #2 (107.9 ± 20.6 ms), athlete #3 (101.6 ± 21.2 ms), athlete #4 (89.1 ± 25.9 ms), and athlete #7 (103.0 ± 48.9 ms). In contrast, athlete #2 and athlete #3 both had a significantly higher RMSSD when compared to athlete #5 (53.4 ± 28.3 ms) and athlete #6 (67.3 ± 21.5 ms).

SDNN was likely lower at week 6 (ES = −0.49), week 7 (ES = −0.42), week 13 (ES = −0.54), week 14 (ES = −0.53), and week 15 (ES = −0.55) (Figure 1D). SDNN was also very likely lower at week 10 (ES = −0.54), week 11 (ES = −0.98), and week 12 (ES = −1.11) compared to baseline (Figure 1D). Like RMSSD, SDNN was very likely higher at week 13 (ES = 0.49), and likely higher at week 14 (ES = 0.50) and week 15 (ES = 0.49) when compared to week 12 (its lowest over the 15 weeks) (Figure 1D).

A significant difference (*p* = 0.04, *n*^2^ = 0.302) in SDNN was reported during the 15-week monitoring period; however, the Bonferroni post hoc test was unable to identify where the difference was. There were also significant differences for SDNN observed between athletes over the monitoring period. Athlete #1 (54.0 ± 21.3 ms) reported significantly lower SDNN when compared to athlete #2 (93.6 ± 16.8 ms), athlete #3 (87.2 ± 16.5 ms), athlete #4 (82.8 ± 19.1 ms), and athlete #7 (98.1 ± 35.2 ms). Likewise, athlete #5 (55.1 ± 26.1 ms) also had a significantly lower SDNN when compared to athlete #2, athlete #3, and athlete #4.

### 3.3. Rowing Performance

Group mean 2000m rowing times were similar (*p* = 0.571) in all three trials (Trial 1—472.1 ± 24.1 s vs. Trial 2—470.1 ± 24.0 s vs. Trial 3—471.2 ± 28.3) for the squad. Each athlete also reported similar 2000 m rowing times between each trial. Expectedly, there were significant differences in 2000 m rowing times between individual athletes.

There were no significant correlations reported between 2000 m ergometer performance and mean HR, RMSSD, SDNN, and training load (all *p* >0.05) at any time point.

## 4. Discussion

The purpose of the current study was to document the weekly fluctuations in cardiac autonomic function in response to varied training load over a winter season (pre-season to start of competitive season), and examine the relationships between training load, heart rate variability, and rowing performance, in competitive female youth rowers. The results from this study reveal a downward trend in HRV estimates of cardiac parasympathetic activity over the course of the season, but not HR, followed by an increase during the final 3 weeks of training as competition approached. In contrast to previous studies, the current study reported no clear relationship between indices of cardiac autonomic activity and training load or rowing performance within this female youth population. Lastly, though measuring resting HR may have value [41], in line with our prior study [27], HRV (RMSSD) may be more sensitive than HR alone in reflecting physiological responses to training in youth athlete populations, whom may have already undergone significant adaptation, as in the present study, changes in HR were less clear. Thus, it might be advisable for youth rowing coaches to monitor HRV, and not just resting HR, to allow for a greater understanding of athlete response to training workload over time and may aid in training planning and management.

### 4.1. Training Load and HRV

The current study shows a downward tendency in HRV-based estimates of cardiac vagal activity (RMSSD, SDNN), but not HR, as the season transitioned from pre-season to in-season training and progressed towards the first competition of the season. Intriguingly, and in line previous research [25], this reduction in RMSSD was not significantly related to changes in training load, and only modest relations were observed. While training load did vary throughout the training season, there was no discernable relationship observed between the variation in training load and the variance in HRV. While this lack of apparent relationship between training load and indices of cardiac autonomic function may be attributed to the way in which training load was quantified in the current study (subjectively by means of RPE), it is also important to note that athletes are subjected to other various stressors outside of training. Given these additional stressors, such as school workload, it is possible that this reduction in vagal modulation was compounded by stressors outside of training, thus limiting the influence of a purely subjective measure of training load on HRV. Few studies have reported the acute effect of psychological stressors on cardiac autonomic function, with a clear link between increased psychological stress and reduced vagal modulation [42,43,44]. However, given the current study did not quantify these potential additional stressors, it is difficult to establish the exact mechanism behind the accumulative reduction in HRV estimated vagal modulation. Likewise, there is also merit in coaches using a more objective measure of training load such as those previously described by Plews and colleagues [25]. Nonetheless, given the previously reported relationship between HRV-based estimates of vagal activity and performance [18,19,31,45], this shift in estimated cardiac autonomic function provides valuable information for coaches working with youth athletes. Future research examining cardiac autonomic function, either directly or indirectly via HRV, in younger athletes, may benefit from additional psychological assessment throughout the monitoring period to ensure a more holistic approach to youth athlete management. Results from the current study (i.e., likely reductions in RMSSD and SDNN) highlight the importance of monitoring stressors external to sport (e.g., psychological, sleep, etc.), in addition to training workload, that might be altering HRV in youth athlete populations, to allow for favorable performance improvements. Further, coaches may be able to tailor team training load based upon alterations in HRV; for example, if athletes, en masse, display significant drops in HRV, the coach may wish to reduce training volume and/or intensity to allow the athletes to better recover and ultimately, avoid maladaptation.

### 4.2. Performance and HRV

As mentioned previously, prior studies have reported a relationship between HRV estimated cardiac vagal modulation and performance in adults [18,19,31,45]. Alternatively, in the current study, no relationships were observed between HRV and rowing performance, with no change in performance times across the three trials in youth athletes. Moreover, results indicate that, despite comparable performance times, RMSSD was reduced at the third rowing trial compared to the first two trials, potentially reflecting a heightened level of pre-time trial anxiety, or anxiety associated with upcoming boat/seat selection and/or proximity to on the water racing. Previous studies have linked pre-competition anxiety to a reduction in vagal modulation [46,47]. A similar reduction was apparent in the third time trial performance in the current study when compared to the first two trials. Interestingly, this reduction in vagal modulation, as assessed by HRV, did not appear to influence performance either negatively or positively, as times across the three trials were similar. While this reduction in HRV-based parasympathetic modulation immediately prior to competition may allow for a greater intensity (i.e., arousal), and as such, enhanced performance [31], performance across the three trials within this athlete population was similar. Alternatively, the lack of relationship between cardiac autonomic activity and rowing performance might be influenced by the time of day [21], though consistent, or incomplete biological maturation as autonomic activity seems to be altered during development [48]. Relatedly, ongoing athletic development (skill improvement) and mental status (e.g., stress, anxiety about boat/seat selection) might also be altering the nature of the relationship between HRV and performance. Further work is needed with larger samples with varied biological and training age, over longer observation periods including more frequent (perhaps daily) assessments of HRV to substantiate these hypotheses. Specific to the unique sport of rowing and potential utility of HRV, future work should explore boat (pair/double, four/quad, or eight) HRV and on the water performance, as rowing is a highly technical sport where high level physical fitness can be amplified or muted by technical proficiency and/or synchronicity of the team.

### 4.3. Experimental Considerations

Participants in the present study were recruited from the largest team in a 100-mile radius (~180 athletes), but to minimize confounding factors, such as sex, age, and training age, coupled with the paucity of studies in females, experienced varsity female rowers were recruited. Due to selectivity factors (experience and sex), the pool of potential participants was 25, and due to the voluntary nature of the study and requiring parental consent, the sample size in the current study was relatively small (n = 7). However, the sample of participants was relatively homogenous and representative of a typical successful youth female varsity (experienced secondary school) rowing squad. Given the success evidenced by either winning or medaling at previous national championship rowing events, the squad was classed as high performing. Further, the statistical approaches employed were chosen to provide alternative assessment beyond the archetypal null hypothesis testing.

## 5. Conclusions

The current study sought to document changes in cardiac autonomic activity over a training season, determine the potential impact of training load on these responses, and ultimately, the impacts on rowing performance. Measures of HRV, but not HR, were found to be reduced over the training season, but were not overtly related to training load or performance. Further work is needed in the youth rowing athlete population to describe HRV-based cardiac autonomic changes over a yearly training program. Youth rowing coaches should consider monitoring HRV of their athletes to potentially modify training magnitude to prevent training maladaptation and ultimately, improve performance.

## Figures and Tables

**Figure 1 ijerph-17-08391-f001:**
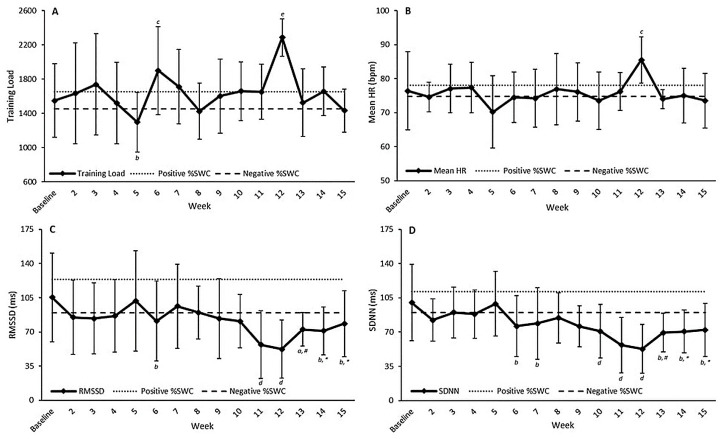
Changes from baseline over the 15-week monitoring period for (**A**) training load, (**B**) mean heart rate, (**C**) root mean square of successive RR interval differences (RMSSD), and (**D**) the standard deviation of all normal to normal RR intervals (SDNN). Data are presented as means ± SD, with dashed lines representing the positive and negative percentage smallest worthwhile change (%SWC, 3%) from baseline [31]. a indicates possibly lower compared to baseline. b indicates likely lower compared to baseline. c indicates likely higher compared to baseline. d indicates very likely lower compared to baseline. e indicates very likely higher compared to baseline. * indicates likely higher compared to week 12. # indicates very likely higher compared to week 12.

**Table 1 ijerph-17-08391-t001:** Individual participant characteristics.

Athlete	Age (years)	Height (cm)	Weight (kg)
1	15	175	74.8
2	17	178	65.8
3	17	175	69.4
4	18	168	71.7
5	16	178	64.8
6	17	183	79.4
7	16	160	47.2
Mean (±sd)	16.6 (±1.0)	173.9 (±7.6)	67.6 (±10.3)

**Table 2 ijerph-17-08391-t002:** Timeline of the HR and HRV monitoring period with 2000 m ergometer trials.

Week														
Baseline	2	3	4	5	6	7	8	9	10	11	12	13	14	15
x	x	x	x	x	x	x	x	x	x	x	x	x	x	x
			2k				2k				2k			

X—HRV recording; 2k—2000 m ergometer trial.

**Table 3 ijerph-17-08391-t003:** Mean values ± standard deviation, variance (95% CI) from baseline, Effect Size (ES), and qualitative inferences for heart rate and heart rate variability during the 15-week monitoring period.

Week															
	Baseline	2	3	4	5	6	7	8	9	10	11	12	13	14	15
Mean HR [bpm]	76.4 ± 11.5	74.6 ± 4.3	77.1 ± 7.3	77.4 ± 7.4	70.2 ± 10.7	74.5 ± 7.4	74.2 ± 8.5	76.9 ± 10.5	76.1 ± 8.6	73.5 ± 8.4	76.2 ± 5.6	85.5 ± 6.8	74.0 ± 2.8	75.0 ± 8.0	73.5 ± 8.0
%△ [95% CI]		−1.5 [−14.4, 13.3]	1.6 [−13.9, 19.9]	1.9 [−13.2, 19.7]	−8.2 [−21.6, 7.7]	−2.0 [−15.1, 13.1]	−2.5 [−14.0, 10.5]	0.8 [−9.2, 12.0]	0.1 [−8.0, 8.9]	−3.4 [−15.0, 9.9]	0.5 [−14.2, 17.7]	12.8 [−5.0, 33.9]	−2.2 [−14.6, 12.1]	−1.4 [−12.5, 11.2]	−3.3 [−16.1, 11.6]
ES		−0.08	0.08	0.10	−0.46	−0.11	−0.14	0.05	0.00	−0.19	0.03	0.66	−0.12	−0.07	−0.18
QI		Unclear	Unclear	Unclear	Unclear	Unclear	Unclear	Unclear	Unclear	Unclear	Unclear	Likely	Unclear	Unclear	Unclear
RMSSD [ms]	105.3 ± 45.4	84.9 ± 38.2	83.8 ± 36.7	86.4 ± 37.1	101.7 ± 51.2	81.2 ± 40.9	96.4 ± 43.1	89.8 ± 26.9	83.6 ± 40.8	80.9 ± 27.4	56.8 ± 34.8	52.4 ± 29.9	72.6 ± 17.3	71.0 ± 24.3	78.4 ± 33.9
%△ [95% CI]		−19.0 [−55.8, 48.5]	−20.9 [−64.0, 73.7]	−18.3 [−65.1, 91.2]	−5.0 [−47.2, 70.8]	−27.0 [−55.0, 18.4]	−10.0 [−52.1, 69.2]	−9.8 [−41.0, 37.8]	−19.8 [−48.0, 23.5]	−20.7 [−58.5, 51.4]	−48.2 [−71.6, −5.4]	−52.7 [−74.4, −12.6]	−25.8 [−55.0, 22.4]	−29.7 [−54.8, 9.3]	−27.6 [−53.8, 13.4]
ES		−0.23	−0.25	−0.22	−0.06	−0.34	−0.11	−0.11	−0.24	−0.25	−0.70	−0.80	−0.32	−0.38	−0.35
QI		Unclear	Unclear	Unclear	Unclear	Likely	Unclear	Unclear	Unclear	Unclear	Very Likely	Very Likely	Possibly	Likely	Likely
SDNN [ms]	100.1 ± 38.9	82.3 ± 21.6	90.1 ± 26.0	88.4 ± 24.7	98.9 ± 33.0	76.0 ± 31.2	78.9 ± 36.7	84.6 ± 25.6	75.9 ± 20.1	70.9 ± 27.4	56.7 ± 28.2	52.8 ± 25.1	69.5 ± 19.6	70.6 ± 22.0	72.2 ± 27.2
%△ [95% CI]		−14.8 [−42.6, 26.4]	−8.0 [−41.4, 44.5]	−9.2 [−49.6, 63.8]	1.4 [−34.9, 58.0]	−25.6 [−46.4, 3.2]	−22.4 [−48.0, 15.6]	−13.7 [−37.2, 18.7]	−21.7 [−49.0, 20.2]	−28.0 [−45.1, −5.5]	−44.9 [−64.9, −13.5]	−48.8 [−69.2, −14.9]	−28.0 [−53.9, 12.5]	−27.6 [−51.3, 7.6]	−28.3 [−49.2, 1.3]
ES		−0.26	−0.14	−0.16	0.02	−0.49	−0.42	−0.24	−0.40	−0.54	−0.98	−1.11	−0.54	−0.53	−0.55
QI		Unclear	Unclear	Unclear	Unclear	Likely	Likely	Unclear	Unclear	Very Likely	Very Likely	Very Likely	Likely	Likely	Likely

HR—heart rate; RMSSD—root mean square of successive differences; SDNN—standard deviation of normal to normal RR intervals; % △—percent variance from baseline; CI—confidence interval; ES—effect size; QI—qualitative inference.

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
