# Peer review of "Heart Rate Variability Responses to a Training Cycle in Female Youth Rowers"

_ijerph, 2020, doi:10.3390/ijerph17228391_

Round 1
Reviewer 1 Report
This study looked at cardiac autonomic modulation, expressed as HRV, on a single day each week over a 15-week mesocycle to examine alterations in parasympathetic modulation in response to weekly training load in addition to seeking a correlation between HRV and a 2000m ergometer time trial. The authors identified a decrease in HRV throughout the 15-week training period without a correlation between training load (srPE) and HRV. Additionally, there were not relationships between physiological measures and time trial performance.
Major Concerns:
- Use of single day HRV measurement for examining weekly exercise stress has been vehemently shown as inaccurate (Plews et al., 2013).
- Current literature examining training load and HRV in rowers is large. As such, in my opinion the novelty of this study and its findings is minimal. Additionally, I believe there is substantial references missing around this area that should be included in the introduction.
- I find the reliance using the SWC (ie. 3%) as established by Buchheit week. Why did the authors not establish this with their own sample? HRV is highly dependent on the individual examined. This should be established to improve the findings.
- When examining HRV for monitoring previous exercise stimuli, it has been proposed morning assessments immediately upon awakening are best suited for athletes (Stanley et al., 2013 & Buchheit 2014).
- Why wasn't individual physiological and performance measures provided? The use of seven participants allows the opportunity to display individual responses and correlations that can provide novel insight beyond the mean response.
Author Response
Reviewer#1
This study looked at cardiac autonomic modulation, expressed as HRV, on a single day each week over a 15-week mesocycle to examine alterations in parasympathetic modulation in response to weekly training load in addition to seeking a correlation between HRV and a 2000m ergometer time trial. The authors identified a decrease in HRV throughout the 15-week training period without a correlation between training load (srPE) and HRV. Additionally, there were not relationships between physiological measures and time trial performance.
Major Concerns:
- Use of single day HRV measurement for examining weekly exercise stress has been vehemently shown as inaccurate (Plews et al., 2013).
Response: While we acknowledge it is ideal to use a daily assessment of HRV, and have mentioned this within the manuscript (page 3, line 111 citing the Plews 2013 paper), work done by Nakamura et al. (2016) and Edmonds et al. (2015) have shown weekly measurements of HRV also offer value. Additionally, since this study was in youth athletes it was simply not feasible to take daily measurements, with the chosen study design reflective of what is practical for a coach or practitioner working with high performing youth athletes. Moreover, as now cited in the paper previous assessment of daily HRV is oft reported as weekly values (Plews et al 2017).
- Current literature examining training load and HRV in rowers is large. As such, in my opinion the novelty of this study and its findings is minimal. Additionally, I believe there is substantial references missing around this area that should be included in the introduction.
Response: We agree that a fair bit of work has been done with heart rate variability and rowers, largely by a single group of respected researchers in this area, though arguably rowing is still less studied than other sports (e.g. soccer). We have revised the introduction to include more references. However, a clear distinction, and thus our contention of novelty, is that this work has almost exclusively been done in males and/or elite level rowers and not females and/or juniors. In fact, this might only be the second paper to publish with this specific population. We have revised this section of the introduction to better make this distinction that female youth rowers are understudied, and particularly with regards to HRV. Thank you for the opportunity to improve our communication of the novelty of this study.
- I find the reliance using the SWC (ie. 3%) as established by Buchheit week. Why did the authors not establish this with their own sample? HRV is highly dependent on the individual examined. This should be established to improve the findings.
Response: Unfortunately, due to time constraints we were not able to obtain multiple measures of the athletes at baseline allowing the calculation of individual level smallest worthwhile change. Though, to the reviewers point, looking at HR for example, using the 0.2 (small effect in Cohen’s d) yields a striking similar cut value 2.3 vs. 2.4 beats/min, for SWC vs. 3%, respectively. Thus, the findings or interpretation thereof are unlikely to be affected by such an overhaul of the analysis, and thus the cost-benefit of re-analysis for a perceptual weakness without much quantitative effect is unfounded.
- When examining HRV for monitoring previous exercise stimuli, it has been proposed morning assessments immediately upon awakening are best suited for athletes (Stanley et al., 2013 & Buchheit 2014).
Response: While we acknowledge that HRV upon awakening is best suited for athletes, we believe a pre-training recording offers useful insight into how a pre-training status affects a youth athletes’ ability to perform. First, it was simply not logistically possible to attain awakening measures of 16-year old athletes in their individual homes before school, let alone on a daily basis. Moreover, this allowed for more complete recovery of HRV, which is suggested to be at least 20 hours at least in a population of highly trained and mostly male rowers (Holt et al., 2019). Further, given youth athletes have to manage the stressors of school and potential work commitments, coupled with their training demands, using a pre-training assessment of HRV offers coaches feedback into how their youth athletes are immediately prior to their training session, rather than a morning assessment that may not truly reflect the added stressors accumulated throughout the athletes day. We believe the study design shows what is practical for a coach or practitioner working with high performing youth athletes.
- Why wasn't individual physiological and performance measures provided? The use of seven participants allows the opportunity to display individual responses and correlations that can provide novel insight beyond the mean response.
Response: Thank you for this suggestion and feedback. Initially, we had chosen to focus our analyses on the group as rowing is a team sport, particularly in the scholastic spring season where there is greater emphasis on the big boats (4 and 8+), when this data was collected. In such team setting, a coach that cannot tailor to an individual, but rather emphasizing the performance of the ‘crew’, thus we think a group analysis is has a practicality to it. However, to the reviewers point, after careful consideration, we have since conducted some individual analyses assessing differences in HR, HRV, training load and rowing performance between athletes over the 15-week monitoring period. The results have been adjusted accordingly.
Reviewer 2 Report
This manuscript evaluated several indices of heart rate variability (HRV) measured once a week during a 15-week training period to estimate the changes in the cardiac autonomic modulation during the training period in seven young female rowers. The findings suggest a decrease in parasympathetic modulation as the training season was getting close to the competition time. The work is interesting and original, and the manuscript is well-written. However, there are several issues that should be addressed, as described below.
Major issues
- The small sample size of the study participants affects the generalization of their findings. The study design with multiple measures over time and the description of consistent methods to decrease bias may increase the internal validity of the study. The reasons for such a small sample are discussed by the authors, but still, it is an important limitation of the study. It is recommended to address the details of the selection process (lines 271 to 276) early on the manuscript (lined 84 to 86).
- The authors describe an approach of the statistical analysis based on magnitude-based decisions. However, it is unclear why an analysis of variance for repeated measures was not performed. Considering the study design, changes within subjects may be relevant to identify the effect of training in the dependent variables (HRV indices), and a combined number of samples increased importantly the total sample size.
- The resolution of Figure 1 and Table 2 should improve significantly.
Minor issues
- Page 2, line 91. Do the authors mean “…based on US rowing junior performance standards” instead of ““…based off US rowing junior performance standards”?
Author Response
Reviewer#2
This manuscript evaluated several indices of heart rate variability (HRV) measured once a week during a 15-week training period to estimate the changes in the cardiac autonomic modulation during the training period in seven young female rowers. The findings suggest a decrease in parasympathetic modulation as the training season was getting close to the competition time. The work is interesting and original, and the manuscript is well-written. However, there are several issues that should be addressed, as described below. paucity
Major issues
- The small sample size of the study participants affects the generalization of their findings. The study design with multiple measures over time and the description of consistent methods to decrease bias may increase the internal validity of the study. The reasons for such a small sample are discussed by the authors, but still, it is an important limitation of the study. It is recommended to address the details of the selection process (lines 271 to 276) early on the manuscript (lined 84 to 86).
Response: Thank you for this suggestion, we have included a more detailed description of the selection process in the Methods section. We also now cite a more recent study of elite male rowers which only included 4 athletes, thus while not ideal and agreeably comes with issue, it is still valuable.
Lines 86-88: “Participants in the study were recruited from the largest team in a 100-mile radius [~180 athletes]. To best minimize confounding factors, such as sex, age, training age, paired with the lack of studies in females, recruitment was limited to experienced varsity female rowers.”
- The authors describe an approach of the statistical analysis based on magnitude-based decisions. However, it is unclear why an analysis of variance for repeated measures was not performed. Considering the study design, changes within subjects may be relevant to identify the effect of training in the dependent variables (HRV indices), and a combined number of samples increased importantly the total sample size.
Response: We sincerely appreciate this suggestion. We have since conducted a repeated measures analysis of variance (ANOVA) to assess differences in HR, HRV, training load and rowing performance over the 15-week monitoring period. The results have been adjusted accordingly to include the ANOVA and appropriate measure of effect size (h2).
- The resolution of Figure 1 and Table 2 should improve significantly.
Response: We apologize for this, we have revised the MS Word settings in an attempt to improve quality.
Minor issues
Page 2, line 91. Do the authors mean “…based on US rowing junior performance standards” instead of ““…based off US rowing junior performance standards”?
Response: Thank you for this suggestion, the manuscript has been revised, and know reads “…within a “high performance” level of rowing ability, when compared against US rowing junior…”.
Reviewer 3 Report
- Line 15 : The purpose of this study on abstract sounds like the methodology, not the objective of the study. It would be great to explain “why is it important to do this work?” and “how this study does impact the sport science?”
- Result section : Please point out some significant weeks and explain the details of them. For example, week 5 training load was dropped, and what were the consequences?
- Figure 1: please explain the stable mean HR of week 6 and 7 while the training load was dropped on week 7.
- Please provide more reasons “how and why does understanding of athlete response to training workload over time and may aid in training planning and management?”. This information will be good on the discussion part.
- I would advise the authors to be more careful concerning the bibliography. It is not consistent. For example,
Plews, D.J.; Laursen, P.B.; Kilding, A.E.; Buchheit, M. Evaluating training adaptation with heart-rate measures: A methodological comparison. International journal of sports physiology 303 and performance 2013, 8, 688-691. – authors used full name of journal
Hautala, A.J.; Kiviniemi, A.M.; Tulppo, M.P. Individual responses to aerobic exercise: The role of the autonomic nervous system. Neurosci. Biobehav. Rev. 2009, 33, 107-115 - authors used abbreviation name of journal
(the DOI may also be mentioned, should the author so desire)
Author Response
Reviewer#3
Line 15: The purpose of this study on abstract sounds like the methodology, not the objective of the study. It would be great to explain “why is it important to do this work?” and “how this study does impact the sport science?”
Response: Thank you for this suggestion, the purpose in the abstract has been revised for clarity and brevity sake.
Result section: Please point out some significant weeks and explain the details of them. For example, week 5 training load was dropped, and what were the consequences?
Response: After another reviewers suggestion we have since conducted, and included, a repeated measures analysis of variance for the data to understand potential weekly effects. As originally included in the manuscript, we conducted correlational analysis on the data and found no such significant associations, but do report significant time effects.
Figure 1: please explain the stable mean HR of week 6 and 7 while the training load was dropped on week 7.
Response: This is an interesting point to pontificate upon, perhaps a physiological lag to the previous weeks overload preventing a drop. However, since we found no such effect between these timepoints this would simply be conjecture.
Please provide more reasons “how and why does understanding of athlete response to training workload over time and may aid in training planning and management?”. This information will be good on the discussion part.
Response: We have added a line to this point in the discussion section of the manuscript (286-289) to this point.
I would advise the authors to be more careful concerning the bibliography. It is not consistent. For example,
Plews, D.J.; Laursen, P.B.; Kilding, A.E.; Buchheit, M. Evaluating training adaptation with heart-rate measures: A methodological comparison. International journal of sports physiology 303 and performance 2013, 8, 688-691. – authors used full name of journal
Hautala, A.J.; Kiviniemi, A.M.; Tulppo, M.P. Individual responses to aerobic exercise: The role of the autonomic nervous system. Neurosci. Biobehav. Rev. 2009, 33, 107-115 - authors used abbreviation name of journal(the DOI may also be mentioned, should the author so desire)
Response: We apologize for such errors. Using reference management software, it can be garbage in-garbage out, meaning a poorly formatted download becomes a poorly formatted reference list. The formatting should now be fixed.
Round 2
Reviewer 1 Report
.
Author Response
We see no comments below to respond to, thus we assume that the reviewer-guided revisions are satisfactory, but perhaps shared concern about the image quality, which is now rectified.
Reviewer 2 Report
The authors addressed all the issues raised by this reviewer.
The resolution of Figure 1 and Table 2 still can improve. In Table 2 it would help to use the template to change to a Word format instead of using an image.
Author Response
We appreciate the efforts of the reviewer in improving the current manuscript. We have since improved the quality of the table (now in MS word) and the figure. Thank you.